# Contribution to the Knowledge of Gastrointestinal Nematodes in Roe Deer (*Capreolus capreolus*) from the Province of León, Spain: An Epidemiological and Molecular Study

**DOI:** 10.3390/ani13193117

**Published:** 2023-10-06

**Authors:** Sara González, María Luisa del Rio, Natividad Díez-Baños, Angélica Martínez, María del Rosario Hidalgo

**Affiliations:** 1Department of Animal Health, Parasitology and Parasitic Diseases, Faculty of Veterinary Science, University of León, 24007 León, Spain; sgonzh00@estudiantes.unileon.es (S.G.); mrhida@unileon.es (M.d.R.H.); 2Department of Animal Health, Section of Immunobiology, Faculty of Veterinary Science, University of León, 24007 León, Spain; m.delrio@unileon.es; 3Department of Biotechnology and Food Science, Faculty of Sciences, University of Burgos, 09001 Burgos, Spain; angelicamd@ubu.es

**Keywords:** gastrointestinal nematodes, prevalence, intensity, species, molecular studies, phylogenetic trees, roe deer

## Abstract

**Simple Summary:**

A study of gastrointestinal nematodes in roe deer was carried out in the regional hunting reserves of Riaño and Mampodre, Province of León, Spain. Through a necropsy, it was observed that all the animals harbored gastrointestinal nematodes in their digestive tract, with the abomasum being the intestinal section where the highest prevalence, intensity of parasitism, and greatest number of identified species were found, followed by the small intestine and the large intestine/cecum. Several of the species found in the study were studied molecularly, and with the sequences obtained compared with those deposited in GenBank, phylogenetic trees were prepared to determine their taxonomic status. The shedding of gastrointestinal nematode eggs in the feces of the animals examined was compared with that of semi-extensive sheep farms in the area. The high values found in the studied parameters show that northern Spain is an area of high-intensity infection for roe deer.

**Abstract:**

A study of gastrointestinal nematodes in roe deer was carried out in the regional hunting reserves of Riaño and Mampodre, Province of León, Spain, to provide information on their prevalence and intensity of infection in relation to the sampling areas, age of the animals, and body weight. Through a regulated necropsy of the animals, all of them harbored gastrointestinal nematodes in their digestive tract, with a mean intensity of parasitism of 638 ± 646.1 nematodes/infected animal. Eleven genera were found and 18 species of gastrointestinal nematodes were identified, three of them polymorphic: *Trichostrongylus axei*, *Trichostrongylus vitrinus*, *Trichostrongylus capricola*, *Trichostrongylus colubriformis*, *Haemonchus contortus*, *Spiculopteragia spiculoptera*/*Spiculopteragia mathevossiani*, *Ostertagia leptospicularis*/*Ostertagia kolchida*, *Ostertagia* (*Grosspiculopteragia*) *occidentalis*, *Teladorsagia circumcincta/Teladorsagia trifurcate*, *Marshallagia marshalli*, *Nematodirus europaeus*, *Cooperia oncophora*, *Capillaria bovis*, *Oesophagostomum venulosum,* and *Trichuris ovis*. All of them have already been cited in roe deer in Europe, but *Marshallagia marshalli*, *Capillaria bovis,* and *Ostertagia* (*Grosspiculopteragia*) *occidentalis* are reported for the first time in Spain in this host. The abomasum was the intestinal section, where the prevalence (98.9%) and mean intensity ($\bar{x}$ = 370.7 ± 374.4 worms/roe deer; range 3–1762) were significantly higher, but no statistically significant differences were found when comparing the sampling areas and age of animals. The animals with lower body weight had a higher parasite load than those in better physical condition, finding, in this case, statistically significant differences (*p* = 0.0020). Seven genera and 14 species were identified. In the small intestine, 88% of the animals examined presented gastrointestinal nematodes, with an average intensity of $\bar{x}$ = 131.7 ± 225.6 parasites/infected animal, ranging between 4–1254 worms. No statistically significant differences were found when the three parameters studied were compared. Four genera and seven species were identified. In the large intestine/cecum, 78.3% of the examined roe deer presented adult worms, with an average intensity of 6.3 ± 5.5 worms/infected animal; range 1–26 worms. Only statistically significant differences were observed when considering the mean intensity of parasitism and the sampling area (*p* = 0.0093). Two genera and two species were identified. Several of the species found in the study were studied molecularly, and with the sequences obtained compared with those deposited in GenBank, phylogenetic trees were prepared to determine their taxonomic status. Using coprological techniques, the existing correlation in the shedding of gastrointestinal nematode eggs in roe deer was investigated with that of semi-extensive sheep farms in the same study area to verify the existence of cross-transmission of these parasites between wild and domestic animals. The high values found in the studied parameters show that northern Spain is an area of high-intensity infection for roe deer.

## 1. Introduction

The roe deer (*Capreolus capreolus*, Linnaeus, 1758) is part of one of the most numerous families of wild ruminants (Cervidae, subfamily Capreolinae) and has the widest distribution in the world. Its presence is limited to the northern hemisphere, and it is widely represented in Europe. In Spain, it is more widely distributed in the north; it has a population of about 600,000 specimens with a 1:2 male/female ratio. It occupies 40% of the national territory with a density of 1–35 roe deer/km^2^, depending on the areas [1]. In our study area, there is a large population of roe deer favored by changes in land use, to the detriment of the surface that was traditionally exploited by extensive livestock. This represents an important resource as a hunting species, with social and economic repercussions linked to the contribution of food and complementary income in traditional local economies.

Roe deer presents a wide variety of endoparasites, particularly gastrointestinal (GI) nematodes, the most common in ruminants worldwide, and several species have been reported in Europe [2,3,4,5,6,7,8,9] and Spain [10,11,12,13,14].

The importance of GI in free-living animals is related to the economic implications for domestic livestock, since they can act as reservoirs of the same and can be cross-transmitted to livestock. This transmission requires a common environment where host species share the same resources, like pasture or watering holes, with consequences for epidemiology, ecology, animal welfare, and economic impact. This is particularly important in areas where unfenced pastures are utilized for transhumance in mountainous regions, such as our study, but also applies to lowland pastures with high densities of roe deer populations crossing fields regularly [15].

The objective of this study was to determine which GI species affect this ungulate in the Province of León, north-western Spain, to show the species that it shares with domestic ruminants in the area when they use the same pastures, to determine the prevalence and intensity of parasitic infection according to their location in the different parts of the digestive system, and thus be able to evaluate the possible health risks caused by these parasites.

Likewise, a molecular study of some of the species found was carried out, and with their sequences, a phylogenetic tree was elaborated to determine their taxonomic status.

## 2. Materials and Methods

### 2.1. Study Area

The study was carried out in the Mountains of Riaño and Mampodre Regional Park (León), coinciding in its northernmost part with the “Picos de Europa” National Park in the Cantabrian Mountains. At the far northeast of the Regional Park is the Riaño Regional Hunting Reserve (43°03′14″ N, 4°57′33.85″ O). It covers an area of 78,000 hectares and represents the Atlantic region in much of its territory. The Reserve was divided into three sampling zones: Queen Land (Zone A), located to the east of the Reserve, with an altitude of 1400 m and bathed by the Yuso River; Oseja and Valdeón Valley (Zone B), located to the north of the Reserve, at an altitude of 1500 m and bathed by the Cares and Sella rivers; and Burón Valley (Zone C) to the west of the Reserve, at an altitude of 1300 m.

To the west, it borders the Mampodre regional hunting reserve (43°01′31″ N, 5°11′18″ W) at the head of the Curueño and Porma rivers. It has an extension of 31,400 hectares and altitudes of more than 2000 m (Figure 1).

In both reserves, the average monthly temperature is 10.2 °C (−12 °C/36 °C). The winter is long and cold, and the summer is short and cool, with an average temperature of 16–17 °C. The rains are distributed throughout the year, with an average annual precipitation of 1088.6 mm, with maximums in October and minimums in July. The landscape is notably mountainous, with plant and animal species characteristic of the habitat. It has a wide and complex fluvial network, as well as numerous tributaries in the form of mountain streams. They are the only places in the Iberian Peninsula where the six most representative species of big game can be shot: red deer, chamois, roe deer, mountain goat, wild boar, and wolf. Wild ruminants share forests and pastures with extensive livestock farming in the area, such as sheep and goats in frank regression and cattle and horses, preferably for meat. In the summer period, livestock in the area increases due to the arrival of transhumant cattle and sheep, mostly from the Community of Extremadura.

### 2.2. Animals Object of Study

The animals were shot by the technical staff of the reserves in selective hunting and transferred to the Faculty of Veterinary Medicine in León. Ninety-two roe deer, all males, were studied: seventy-six from the three sampling areas into which Riaño Regional Hunting Reserve was divided, and sixteen from Mampodre Regional Hunting Reserve.

The roe deer sampled were classified into two age groups, young ≤ 2 years old (n = 32) and adults > 2 years old (n = 60), and they were also classified according to their body weight (Table 1).

At the same time, feces samples were collected from 151 semi-extensive sheep farms in the areas where the roe deer were obtained. From each farm, a fecal mixture of 5% of the sheep (females between 4 and 7 years of age) was collected.

### 2.3. Sampling Procedures

Upon arrival of the animals at the laboratory, they were identified, and data regarding the place of capture, age, weight, and other biometric measurements were taken.

A regulated necropsy was performed, and after accessing the abdominal cavity through the linea alba, the gastrointestinal tract was extracted. The abomasum, the small intestine, the large intestine, and the cecum were collected separately. The content of each part was washed with warm water under pressure to thoroughly detach all the worms from the gastric walls and was filtered through three sieves of 500, 320, and 150 µm pore diameter. The filtered material was collected in one-liter sedimentation cups. 90% of the filtered material was taken and stored in 5% formalin for the identification and quantification of nematodes. The rest of the filtered material (10%) was preserved in 70% ethanol for its molecular study.

### 2.4. Morphological Identification of Nematodes

The formalin-preserved material was examined in Petri dishes under a stereoscopic microscope (Nikon SMZ 1500, Tokyo, Japan), and, with the aid of an entomological needle, all nematodes were collected. Subsequently, they were counted, separated by sex, and mounted in semi-permanent preparations of 0.05% lactophenol cotton blue stain. They were studied using a light microscope (Nikon YS-CF 100, Tokyo, Japan) [16,17,18,19,20,21,22,23,24,25] and photographed using a Leitz Wetzlar light microscope and a microscope equipped with Nomarski lenses (NIKON DS-Fi1). For scanning electron microscopy observation (SEM, JEOL, JSM-6480-LV, Tokyo, Japan), the parasites were fixed with 2.5% glutaraldehyde, post-fixed with 2% osmium tetroxide, rinsed with PBS, and dehydrated in aqueous ethanol at increasing concentrations for 30 min per step (30%, 50%, 70%, 90%, 3 × 96%, and 3 x 100%). Afterwards, the samples were dried with carbon dioxide and mounted on cylindrical metal supports, coated with conductive silver glue, and then metalized with gold, obtaining digital images at different magnifications.

### 2.5. Molecular Analysis

The material preserved in 70% ethanol was used for the molecular study. The nematodes were extracted and divided into two portions: the anterior end was used for DNA extraction, and the posterior part was used for the identification of each specimen.

For DNA extraction, a lysis solution consisting of 80 mL of TrisHCl (50 mM), 10 mL of KCl (1 M), 6 mL of MgCl2 (50 mM), 1 mL of Tween 20, 950 µL of Nonidet P40, and 0.02 g of Gelatin was used. Make up to 200 mL with MiliQ water and autoclave at 121 °C for 15–20 min.

The DNA extraction protocol was as follows:

The anterior end of the worm, previously identified, was introduced into a 1.5 mL Eppendorf, and 100 µL of lysis buffer and 5 µL of Proteinase K (25 mg/mL) were added. After shaking, it was incubated at 56 °C for 50 min until the tissue was completely lysed. Subsequently, Proteinase K was inactivated at 98 °C for 10 min, and the sample was centrifuged at 4 °C and 14,000 rpm for 10 min. Finally, 60 μl of the supernatant was taken for PCR amplification.

The second internal transcriber spacer region (ITS-2) was amplified by PCR using the primers NC1 (5′-ACGTCTGGTTCAGGGTTGTT-3′) and NC2 (5′-TTAGTTTCTTTTCCTCCGCT-3′) [26], for the genus *Spiculopteragia* [27], *Ostertagia* [28], *Trichostrongylus* [29], and *Nematodirus* [30].

Purified PCR products were sequenced using a Thermosequenase cycle sequencing kit in the sequencing DNA core facility of the University of León. Reference sequences obtained from Genbank corresponding to *Trichostrongylus axei* (EF427622), *Nematodirus europaeus* (AJ239114), *Oesophagostomum venulosum* (HQ283349), *Spiculopteragia spiculoptera* (DQ354329), *S. mathevossiani* (DQ354328), Hembras *Spiculopteragia spiculoptera* (DQ354329), *Ostertagia leptospicularis* (DQ354331), *Ostertagia kolchida* (DQ354333), and Hembras *Ostertagia leptospicularis* (DQ354333) were compared and aligned with the nucleotide sequences obtained from our amplified samples using Clustal W [31]. At the same time, the composition of G + C (%) of the sequence was calculated, and the phylogenetic trees were made using the Neighbor-Joining algorithm (grouping method) and the Kimura 2-parameter model to perform the phylogeny of the species involved.

### 2.6. Coprological Analysis

Feces samples were taken directly from the rectum of the animals in order to avoid contamination and, after being labeled and identified, were immediately processed using the McMaster flotation technique with a saturated sodium chloride solution (1.2 density) [32]. The samples were examined with a light microscope to detect and quantify GI nematode eggs and coccidia oocysts.

### 2.7. Statistical Analysis

The prevalence and mean intensity of the infection were determined, considering age and study area [33]. Data analysis was carried out using the chi-square non-parametric test in order to compare the prevalence, as well as the Mann–Whitney and Kruskal–Wallis tests to compare the intensity. The 95% confidence intervals (CI) for prevalence and mean intensity were calculated [34]. The median intensity and the aggregation index (variance/mean ratio) were calculated in order to observe the distribution of nematodes in their hosts [35].

## 3. Results

### 3.1. Necropsy Results

The prevalence of nematodes in the gastrointestinal tract of roe deer examined in the present study was 100%, with a mean intensity of 491.7 ± 481, 95% CI: 450.7–533.3; range 9–2755 nematodes/infected animal. A total of 45,235 nematodes were collected, of which 66.9% were female and 33.1% were male (female/male ratio = 2.02). The parasite loads were moderate; 38% of the animals had values higher than 500 worms, and in two young animals, it was higher than 2000 nematodes/roe deer. These data indicate that the distribution of the nematodes in the infested animals showed an aggregated/overdispersed pattern (median intensity = 334.5; aggregation index = 470.2), where the nematodes are not randomly or uniformly distributed, but strongly aggregated among their hosts.

Regarding the mean intensity of parasitism, it was observed that young animals were more parasitized ($\bar{x}$ = 638 ± 646.1, 95% CI: 587.8–688.2 worms/roe deer; range 9–2755 nematodes/animal) than adults ($\bar{x}$ = 413.7 ± 346, 95% CI: 447.3–380.7, range 33–1413), and animals with lower body weight (≤22 kg) had a higher parasite load ($\bar{x}$ = 690.9 ± 597.1, 95% CI: 646.5–735.5; range 63–2755) than those of greater body weight, whose mean intensity did not exceed 415 nematodes/animal, observing statistically significant differences only for this parameter (*p* = 0.0105).

In the four sampling areas, all the animals had nematodes in their gastrointestinal tract, but it was the roe deer from the Mampodre Reserve that presented the lowest parasitic intensity ($\bar{x}$ = 375.4 ± 340.8, 95% CI: 341–410 nematodes/roe deer; range 9–1095 nematodes/animal), while in the three areas of the Riaño Reserve, the variability in the parasite load was wider, although no statistically significant differences were observed (*p* = 0.8451). (Table 2).

Eleven genera were found, and 18 species of GI nematodes were identified, 3 of them polymorphic:

*Trichostrongylus axei* (Cobbold, 1879) Railliet & Henry, 1909

*Trichostrongylus vitrinus* Loss, 1905

*Trichostrongylus capricola* Ransom, 1907

*Trichostrongylus colubriformis* (Giles,1892) Ransom, 1911

*Haemonchus contortus* (Rudolphi, 1803) Cobb, 1898

*Spiculopteragia spiculoptera* (Guschanskaja, 1931) Orloff, 1933/*Spiculopteragia*

*mathevossiani* Rukhliaadev, 1848

*Ostertagia leptospicularis* Assadov, 1953/*Ostertagia kolchida* Popova, 1937

*Ostertagia* (*Grosspiculopteragia*) *occidentalis* Ransom, 1907

*Teladorsagia circumcincta* (Stadelmann, 1894) Drozdz, 1965*/Teladorsagia trifurcata*

(Ransom, 1907) Drozdz, 1965

*Marshallagia marshalli* (Ransom, 1907) Orloff, 1933

*Nematodirus europaeus* Jansen, 1972

*Cooperia oncophora* (Raillet, 1898) Ransom, 1907

*Capillaria bovis* Schnyder, 1906

*Oesophagostomum venulosum* Rudolphi, 1809

*Trichuris ovis* Abilgaard, 1795.

All of them have already been cited in roe deer in Europe, but *Marshallagia marshalli*, *Capillaria bovis,* and *Ostertagia* (*Grosspiculopteragia*) *occidentalis* are reported in this host for the first time in Spain. Similarly, these species have been cited in sheep, except for *Nematodirus europaeus* and the morphotypes *Spiculoptera spiculoptera*/*S. mathevossiani* and *Ostertagia leptospicularis*/*O. kolchida*. Its morphological characteristics can be seen in Figure 2, Figure 3, Figure 4, Figure 5 and Figure 6.

#### 3.1.1. Abomasum Parasitism

The abomasum was the part of the gastrointestinal tract where the highest prevalence, highest number of nematodes, and diversity of species were found. In this study, 98.9% (95% CI: 96.8–100%) of the animals presented parasites in this location, with moderate mean intensity ($\bar{x}$ = 370.7 ± 374.4, 95% CI: 331.8–408.2 worms/roe deer; range 3–1762), and 75% of the total parasites found in this study (34108 nematodes) were collected. The female/male sex ratio was 1.86.

In this gastric section, the parasite load was lower in the animals from the Mampodre Reserve ($\bar{x}$ = 318.2 ± 322.8, 95% CI: 282.7–353.7 nematodes/roe deer, range 3–1091) compared to those sampled in the three areas of the Riaño Reserve, and despite this, no statistically significant differences were found when comparing the sampling areas (*p* = 0.9580). Young animals harbored a mean of 465.7 ± 448.3, 95% CI: 425–506.4 nematodes/roe deer (range 3–1762), higher than that found in adult animals ($\bar{x}$ = 320.1 ± 321.1, 95% CI: 284.8–355.2, range 21–1373), but no significant differences were observed (*p* = 0.1775). The animals with lower body weight (≤22 Kg) had a higher parasite load than those in better physical condition, finding, in this case, statistically significant differences (*p* = 0.0020) (Table 3).

Seven genera and 14 species were identified, 3 of them polymorphic. Nematodes of the subfamily Ostertaginae belonging to the genera *Spiculopteragia*, *Ostertagia*, *Teladorsagia,* and *Marshallagia* were the most abundant (96.7%, 95% CI: 93.1–100%; $\bar{x}$ = 205.7 ± 243.2, 95% CI: 172.8–239.2), followed by the genus *Trichostrongylus* (84.8%, 95% CI: 77.5–92.1%; $\bar{x}$ = 203.4 ± 273.9, 95% CI: 179–227), and to a lesser extent, *Haemonchus* (16.3%, 95% CI: 8.7–23.8%) and *Nematodirus* (7.6%, 95% CI: 2.2–13%).

Multispecific infections were always found, with triple (35.9%) and quadruple (33.7%) infections predominating, with the genera *Trichostrongylus*, *Ostertagia*, *Spiculopteragia,* or *Teladorsagia* being involved, followed by quintuple (15.2%) and double (9.8%) infections. The least represented were simple (4.4%) and sextuple (1.19%) infections.

The nematodes of the subfamily Ostertaginae represented 53.68% of the worms found in the abomasum and 40.56% of the total parasites found in the roe deer sampled. The majority were females (10730) and presented a mean intensity of $\bar{x}$ = 120.6 ± 136.4, 95% CI: 96.3–145; range 2–698 females/parasitized animals. The sex ratio was 1.41. Due to the difficulty of their morphological identification, they were divided into females with (39%) or without vulvar flaps (61%).

In our study, the presence of three polymorphic species was observed. *T. circumcincta*/*T. trifurcata* presented the highest values of both prevalence and intensity of infection (86.9%, 95% CI: 80–93.8%; $\bar{x}$ = 47.6 ± 67.9, 95% CI: 28.3–67; 1–400), followed by *S. spiculoptera*/*S. mathevossiani* with a lower prevalence (77.2%, 95% CI: 68.6–85.8%) and intensity of parasitization ($\bar{x}$ = 42.9 ± 62.6, 95% CI: 24.2–61.6 worms/infected roe deer; range 1–365) and finally, *O. leptospicularis*/*O. kolchida* (53.3%, 95% CI: 43.1–63.5%; $\bar{x}$ = 11.6 ± 14.9, 95% CI: 36–20.2; range 1–84) (Table 3).

*Ostertagia* (*Grosspiculopteragia*) *occidentalis* has been reported for the first time in this host on the Iberian Peninsula. It was only found in three adult animals and one young, from zone B of the Riaño Reserve, with a low prevalence (4.3%, 95% CI: 0.2–8.5%) and mean intensity ($\bar{x}$ = 3.2 ± 2.1, 95% CI: 1–5.5; range 1–6 nematodes/infected roe deer).

Like the previous species, *Marshallagia marshalli* is the first record in Spain for roe deer. It was identified in 22 adult animals and in 7 young, being in Zone B of the Riaño Reserve, where its highest representation was found (46.4%, 95% CI: 36.2–56.6%; $\bar{x}$ = 8.6 ± 8.8, 95% CI: 2.8–14.4; range 1–35).

In the gastric tract of roe deer, four species of *Trichostrongylus* were found, the most frequent and abundant being *T. axei* (84.8%, 95% CI: 77.5–92.1%; $\bar{x}$ = 52.9 ± 74.2, 95% CI: 33–73 nematodes/infected roe deer; range 9–336 worms). Lower values were observed for *T. vitrinus* (34.8%, 95% CI: 25.1–44.5% $\bar{x}$ = 4 ± 4.7, 95% CI: 0.1–8;1–27), *T. capricola,* and *T. colubriformis* (Table 3). The highest parasite load of *T. axei* was found in young animals ($\bar{x}$ = 56.4 ± 84.8, 95% CI: 62.7–107, range 1–276) and in the Riaño Reserve ($\bar{x}$ = 55.2 ± 76.6, 95% CI: 35–75.4, range 1–175).

#### 3.1.2. Small Intestine Parasitism

Parasites were identified in 88% (95% CI: 81.3–94.7%) of the small intestines examined, with an average intensity of $\bar{x}$ = 131.7 ± 225.6, 95% CI: 100–163 nematodes/infected animal, ranging between 4 and 1254 worms. The female/male sex ratio was 2.64. A total of 10672 worms were collected (23.6% of the total number of worms found), and most of them were identified as *Nematodirus europaeus*.

Young animals presented a lower prevalence than adult animals, but with a slightly higher mean intensity of infection ($\bar{x}$ = 223.7 ± 362.8, 95% CI: 176.2–271.2, range 1–1254), not finding statistically significant differences (*p* = 0.5014), not even when related to the body weight of the animals (*p* = 0.1970). The least parasitized roe deer were observed in the Mampodre Reserve ($\bar{x}$ = 75.2 ± 59.9 95% CI: 61.7–88.8 worms/infected animal), but without statistically significant differences when considering the different sampling areas (*p* = 0.9380) (Table 3).

In the small intestine, four genera and seven species were identified: *Nematodirus europaeus*, *Trichostrongylus axei*, *Trichostrongylus vitrinus*, *Trichostrongylus capricola*, *Trichostrongylus colubriformis*, *Capillaria bovis,* and *Cooperia oncophora*, all of them with variable prevalences, but with low mean intensities. Double infections by *N. europaeus* and *T. axei* (48.1%) predominated, followed by triple infections (30.9%), simple infections by *N. europaeus* (13.6%), and, lastly, quadruple infections (7.4%).

*Nematodirus europaeus* was identified in 80.3%, 95% CI: 72.3–88.5% of the roe deer examined, with a mean intensity of $\bar{x}$ = 115.3 ± 221.7, 95% CI: 74.8–155.7 worms/infected animal and a range between 1 and 1226 nematodes/roe deer, and represented 79.9% of the total number of worms found in this location and 18.8% of the total number of nematodes observed. The prevalence was higher in adult animals (95%), but the highest mean intensity of parasitism was found in young animals ($\bar{x}$ = 223.7 ± 362.8, 95% CI: 176.2–271.2 nematodes/infected roe deer; range 4–1254).

The genus *Trichostrongylus* was found in the small intestine in 72.8%, 95% CI: 63.7–81.9% of the roe deer, with a mean intensity of 24.4 ± 32.8, 95% CI: 11.4–37.4 worms/roe deer, ranging from 5–173 worms/animal. As in the abomasum, four species were identified. *T. axei* had the highest prevalence and mean intensity (51.2%, 95% CI: 41–61.4%; $\bar{x}$ = 5.5 ± 5.5, 95% CI: 1–10 nematodes/infected roe deer; range 1–46 worms), while *T. vitrinus* and *T. capricola* had lower values. *T. colubriformis* was only identified in seven roe deer, all of them from the Riaño Reserve. In young animals, both the prevalence and the parasite load of this genus were low and similar to that found in adult roe deer, except in adult roe deer from zone C that presented a slightly higher mean intensity, but no statistically significant differences were found (Table 3).

*Capillaria bovis* and *Cooperia oncophora* were identified in 35.9%, 95% CI: 26.1–45.7%, and 17.4%, 95% CI: 9.7–25.1%, respectively, of the sampled roe deer, which did not exceed the mean of 5 worms per animal and represented less than 1.5% of the worms found in this intestinal section (Table 3).

#### 3.1.3. Large Intestine/Cecum Parasitism

In this location, 78.3%, 95% CI: 70–86.7%, of the examined roe deer presented adult worms, with an average intensity of 6.3 ± 5.5, 95% CI: 2–10.6 worms/infected animal; range 1–26 worms. 465 worms were collected. The female/male ratio was 2.47.

The prevalence in adult animals was higher (83.3%, 95% CI: 75.7–90.9%) than in young animals (75%, 95% CI: 72.9–77%), presenting similar parasite loads ($\bar{x}$ = 6.4 ± 5.1; 95% CI: 2.5–25.6) and $\bar{x}$ = 5.9 ± 6.2; 95% CI: 1–11 nematodes/animal, respectively), without statistically significant differences (*p* = 0.4030). Regarding the sampling areas, in the Mampodre Reserve the lowest number of infected animals and the lowest parasite load were found, while in Zone B of the Riaño Reserve all the roe deer were parasitized ($\bar{x}$ = 5.0 ± 5.6; 95% CI: 0.2–10 nematodes/animal). Statistically significant differences were observed when considering the mean intensity of parasitism and the sampling area (*p* = 0.0093). (Table 2).

Two species were identified. *Oesophagostomum venulosum* was observed in 58.7%, 95% CI: 48.6–68.7% of the animals, and *Trichuris ovis* in 64.1%, 95% CI: 54.2–73.8%, with parasite loads that did not exceed the mean of 2.5 worms per animal. Double infections (55.5%) prevailed over simple ones.

### 3.2. Coprological Data

In roe deer, the excretion of trichostrongylid eggs was quite high in terms of prevalence but very low with respect to parasitic intensity (65.2%; $\bar{x}$ = 47.2 ± 46.8 hgh, range 6–260 epg), with variable data regarding the presence of eggs of *Trichuris*, *Moniezia,* and coccidia oocysts (Table 4).

In the feces samples analyzed from the semi-extensive sheep farms located in the surroundings of the Riaño and Mampodre hunting reserves, trichostrongylid eggs were observed in 87.7% of the farms, but with moderate mean intensities ($\bar{x}$ = 143 ± 233) ranging between 13–1.700 eggs per gram of feces (epg), and the presence of *Nematodirus*, *Monieza,* and *Trichuris* was almost occasional with respect to the number of samples examined. Higher prevalences (96%) were observed in the elimination of coccidia oocysts (Table 4).

### 3.3. Molecular Results: Sequences

#### 3.3.1. DNA Sequences of *Spiculopteragia spiculoptera*/*S. mathevossiani*

Five male specimens of the polymorphic species *S. spiculoptera*/*S. mathevossiani* were analyzed; three of them were identified as the majority morphotype, *S. spiculoptera*, and the other two as the minority morphotype, *S. mathevossiani*. The sequence length was 229 bp. The five sequences were compared with the one deposited in GenBank obtained from isolates of *Capreolus capreolus* (DQ354328). It is confirmed that both morphotypes constitute a single species: *S. spiculoptera*/*S. mathevossiani*. The sequences were deposited in GenBank with the accession numbers OR257709, OR257710, OR257711, OR257712, OR257713 and the name of the majority morphotype. The % G+C of the sequences was 37%.

#### 3.3.2. DNA Sequences of *Ostertagia leptospicularis*/*O. kolchida*

Five male specimens of the polymorphic species *O. leptospicularis*/*O. kolchida* were molecularly analyzed; two of them were identified as the majority morphotype, *O. leptospicularis*, and the other three as the minority morphotype, *O. kolchida*. The sequence length was 224 bp. The five sequences were compared with those deposited in GenBank, obtained from isolates of *Capreolus capreolus* (DQ 354333 and DQ354331). It is confirmed that both morphotypes constitute a single species: *O. leptospicularis*/*O. kolchida*. The sequences were deposited in GenBank with the accession numbers OR257701, OR257702, OR257703, OR257704, and OR257705 and the name of the majority morphotype. The % G+C of the sequences was 33.2%.

#### 3.3.3. DNA Sequences of Females of the Ostertagiinae Subfamily

Six females of this subfamily were molecularly studied; three females presented vulvar flaps, and the other three lacked this characteristic morphological element.

The sequence length of the females with vulvar flap was 222 bp and was compared with one deposited in GenBank, obtained from isolates of *Capreolus capreolus* (DQ354333) for *O. leptospicularis*/*O. kolchida*. The sequences were deposited in GenBank with the accession numbers OR257706, OR257707, and OR257708. The % G+C of the sequences is 34.8%.

The sequence length of the females without vulvar flap was 224 bp. coinciding with that deposited in GenBank, DQ354329, for *S. spiculoptera/S. mathevossiani*. The sequences were deposited in GenBank with the accession numbers OR257714, OR257715, and OR257716. The % G+C of the sequences is 36.9%.

#### 3.3.4. DNA Sequence of *Trichostrongylus axei*

The DNA of a male worm identified morphologically as *T. axei* was studied. The length of the sequence was 223 base pairs (bp) and agrees with that deposited in GenBank for this species, obtained in sheep in Russia, EF427622. The sequence was deposited in Genbank with the accession number OR264572. The %G+C of the sequence was 34.5%.

#### 3.3.5. DNA Sequence of *Nematodirus europaeus*

A female morphologically identified as *N. europaeus* was molecularly studied. The sequence length was 207 bp and was deposited in GenBank with the accession number OR257700. This sequence was compared with that deposited in roe deer in Italy (AJ239114). The %G+C of the sequence was 45.7%.

#### 3.3.6. DNA Sequence of *Oesophagostomum venulosum*

Two worms were studied, one female and one male, morphologically identified as *O. venulosum*. The sequence length was 528 bp and was compared with that found in sheep and deposited in GenBank with the accession number HQ283349. The sequences obtained in our study were deposited in Genbank with the accession numbers OR255925 and OR255926. The % G+C of the sequences was 45.9%.

### 3.4. Phylogenetic Trees

The sequences of the ITS-2 region of the rDNA of different species of gastrointestinal nematodes introduced by various authors were obtained from GenBank and carried out with them. Together with our results, a phylogenetic tree shows the positions of the different species of *Spiculopteragia*, *Ostertagia,* and *Teladorsagia* and another tree shows the positions of *Nematodirus europaeus*, *Trichostrongylus axei,* and *Oesophagostomum venulosum* and related species (Figure 7 and Figure 8).

## 4. Discussion

The knowledge of nematode communities in wild hosts is currently a fact of general biological interest from a veterinary and public health point of view.

According to the Iberian Zooparasite Index, there are some 30 species of GI nematodes described in Spain in ruminants [43], some of which are host-specific, and many of those that affect wild ruminants are shared by domestic livestock [44]. Of them, *Teladorsagia circumcincta*, *Trichostrongylus axei*, *T. vitrinus,* and *T. colubriformis* are considered zoonotic [45]. In the present study, a high number of genera (11) and species (18) of GI nematodes in roe deer have been identified, all of them cited in this host in Spain and other European countries [4,5,6,7,10,11,14]. The great diversity of species found may be explained by the behavior of the roe deer. They live in forests, moving long distances, grazing in open areas with other ruminants, wild or domestic, and approaching humans in agricultural and even urban areas.

Helminth infection in the roe deer examined was prevalent (100%) with a moderate mean intensity (491.7 ± 481 nematodes/deer infected). This high prevalence and moderate intensity of infection is in accordance with several studies for these parasites in wild cervids [4,5,7,14,46,47,48], which confirms that they are widespread infections in Europe and Spain. These results will depend on the ecological conditions, favorable or not, for the development of the free phases of the life cycle of the parasites, which will be different depending on the geographical area studied.

Among the different segments of the GI tract, the highest prevalence, intensity, and variety of species were found in the abomasum, coinciding with previous work [4,5,7,8,14,47,48,49]. The mean infection intensity was different depending on the geographical area, species found, and parameters analyzed. Thus, when the parasite load was related to the weight of the animals, statistically significant differences were observed, and some authors observed, as in our study, that roe deer with poor body condition had a higher parasite load [5]. These animals with lower body weight generally corresponded to cachectic adult animals with signs of malnutrition, malabsorption of nutrients, and a depressed immune system, making them more susceptible to acquiring any infectious or parasitic process. In young animals, lower weight would be related to their smaller size but not to poor body condition. In our study, statistically significant differences were detected when considering this parameter.

When considering the age of the animals, we observed that the young were more parasitized than the adults, as was the case with other authors [14], although on other occasions the adults were the most parasitized [2]. However, no statistically significant differences were observed with this factor. Being young animals, they would be indicating to us that they have not yet developed a complete protective immune response, so they would be more parasitized, while this response in adult animals would be more developed.

The most frequent species found in the abomasum were *T. circumcincta*/*T. trifurcata*, *S. spiculoptera*/*S. mathevossiani,* and *O. leptospicularis*/*O. kolchida*. The last two are considered by a large number of authors to be the most widely represented in roe deer in Europe [2,4,6,50,51]. Both are considered primary pathogens of wild ruminants [27,52] and are occasionally cited in domestic animals [53,54]. The different species of the genus *Teladorsagia* are primarily parasitic species of domestic animals [52], mainly of the Bovidae. Although it was found with a high prevalence, it is not a frequent species in this host and has hardly been mentioned in other studies.

*Haemonchus contortus* is considered the most important pathogenic parasitic species affecting sheep. On a few occasions, our study found a low prevalence and intensity of data that coincided with those observed in the Iberian Peninsula [14] and different European areas [2,4]. However, in other areas of Italy [6], Ukraine [7], Austria [15], Sweden [55], and Turkey [51], it was one of the most prevalent parasitic species of roe deer, whose high intensities can cause serious clinical signs of haemonchosis and even the death of the animals.

*Ostertagia (Grosspiculopteragia) occidentalis* was found with a low prevalence and in a low number of adult animals examined. This is the first time that this species has been cited in Spain in this host, as well as *Marshallagia marshalli*, a species cited in roe deer in Italy with low prevalence [4,6]. In Spain, both parasites are more frequent in sheep and goats [56].

We have found four species of the genus *Trichostrongylus* present in the abomasum and small intestine, although with a generally higher prevalence in the first location: *T. axei* (84.8–51.2%), *T. vitrinus* (34.8–23.9%), *T. capricola* (16.3–20.6%), and *T. colubriformis* (9.8–9.8%). All of them are mainly parasitic species of sheep, goats, and cows (except *T. capricola,* which has not been reported in cattle) and have been cited in areas with different climatic conditions, such as Italy [6], Russia [8], Spain [14], or Turkey [51].

Multispecific infections were always observed in this location, with triple and quadruple infections predominating and being more harmful than monospecific ones, as different pathogenic mechanisms were involved [51].

In the small intestine, seven species were identified. In addition to the four species of *Trichostrongylus* already mentioned, the one that predominated in this intestinal tract was *Nematodirus europaeus*, considered specific for this host. It has been found by several authors to be the most prevalent in this wild ungulate [2,4,50]. Previous studies carried out in Spain cite it as *N. filicollis* [10,11,49], although other authors consider it a synonym of *N. europaeus* [57]. In this study, we confirm that the molecular sequence analyzed for this species matches *N. europaeus* when compared to that deposited in GenBank, AJ239114, obtained from isolates of roe deer in Italy [41].

The presence in the small intestine of *Cooperia oncophora* has been sporadic since it is a parasite of cattle and other ruminants [58,59]. The same happens with *Capillaria bovis.* It has been previously confirmed in the north of Spain in deer [59], and its presence has been confirmed in cattle, the usual host of the species [60,61].

Two species were identified in the large intestine and cecum: *Oesophagostomum venulosum* and *Trichuris ovis,* presenting high prevalence but very low mean intensities. Likewise, several authors [2,5,51] have pointed out a high prevalence of these two species in European roe deer. It is the intestinal segment where the lowest prevalence and mean intensity of infection were observed, as has already been stated by several authors, and where double infections predominate, as in the small intestine.

As we can see, parasitic gastroenteritis is very widespread parasitosis and is present in both domestic and wild animals, which can contribute to the cross-transmission of parasites between them by sharing the same habitat, in addition to the possible reservoir role that wild animals would have. All these diseases have a negative effect on the energy reserves of the animals, resulting in loss of vitality, nutritional status, and growth, which can adversely affect the survival capacity of the affected animals. In some studies, it has been determined that parasitic diseases constitute 11–48% of roe deer mortality [9,62]. All this suggests the need for new research in order to determine the health status of these populations and be able to establish preventive measures for parasite control.

The morphological identification of GI nematodes is complex since it is based on characteristics that are sometimes difficult to perceive, such as the morphology and length of the spicules and the genital cone in males or the position of the synlophyum in females. The molecular study guarantees us a precise taxonomy that is fundamental for the study of potentially pathogenic parasite species [27], especially in those species of the Ostertaginae subfamily where polymorphic males have been observed.

Thus, regarding the molecular study, in our work, three polymorphic species were found. Only two of them have been studied molecularly. The polymorphism has been verified in the Trichostrongylidae, Cooperiidae, and Haemonchidae families, being frequent both in wild and domestic ruminants. It has been confirmed, among other species, in *Ostertagia leptospicularis*/*O. kolchida*. Before being able to demonstrate whether two species were genetically identical, the hybridization of *Ostertagia leptospicularis* and *O. kolchida* was studied, demonstrating that both constituted a polymorphic species [53,63]. Subsequently [27], the ITS-2 region of *Ostertagia leptospicularis* and *O. kolchida* was studied, which showed that *Ostertagia leptospicularis* and *O. kolchida* constituted a single species and confirmed the previously proposed theory [53,63]. In our study, we also confirmed that both morphotypes constitute a single species, and the sequences obtained were deposited in GenBank under the name *Ostertagia leptospicularis*.

Females of the Ostertagiinae subfamily are difficult to differentiate morphologically. Due to this reason, there are few genetic studies carried out. In the majority, males were used because the characteristics of the copulatory bursa and the spicules allow a better taxonomic classification. We only differentiate between females with and without vulvar flaps. In the genetic tests of the females that presented vulvar flap, their sequences were identical to those of *Ostertagia letospicularis*/*O. kolchida*, and those without it matched the sequences of *S. spiculoptera*/*S. mathevossiani*. However, we cannot state that females of the genus *Spiculopteragia* lack a vulvar flap, since females of *S. heudemeri*/*S. andreevae* were observed with this characteristic [64].

These results are probably due to the low number of samples analyzed, so more studies are required. The study of the ITS-2 region in females, as in males, is adequate for the taxonomic study of these two species. Using the ITS-1 region and mitochondrial DNA from *Ostertagia* females, the same results were obtained [65].

The other polymorphic species studied molecularly was *Spiculopteragia spiculoptera*/*S. mathevossiani*. Like *Ostertagia leptospicularis*/*O. kolchida*, there is a good review of *S. spiculoptera*/*S. mathevossiani* [27]. The five sequences obtained in our study—three of them identified as the majority morphotype, *S. spiculoptera*, and the other two as the minority morphotype, *S. mathevossiani*—were identical, so our results coincide with those of other authors [27], who stated that *S. spiculoptera* and *S. mathevossiani* are genetically identical and constitute a single species.

A male morphologically identified as *T. axei* was studied molecularly. The sequence obtained coincided with the one deposited in GenBank (EF427622). The ITS-2 region is considered to be specific in the taxonomic and phylogenetic study of this species, regardless of the host in which they have been found [29,40], since these parasites affect a wide range of hosts [43].

In the small intestine of the examined roe deer, *N. europaeus* was found in great proportion, and one female was molecularly studied. Her sequence was compared with the one deposited in GenBank [41], confirming its identity. The morphological identification of the species is difficult, as it can easily be confused with *N. filicollis*. For this reason, the same author [41] genetically related four species of *Nematodirus* present in wildlife (*N. rupicaprae*, *N. davtiani alpinus*, *N. europaeus,* and *N. oiratianus*), which are phylogenetically close, but distant from the species that affect domestic ruminants (*N. spathiger*, *N. filicollis,* or *N. battus*). For this reason, they affirmed that its molecular identification is necessary in order not to confuse the species present in wild animals with those of domestic ruminants.

The second internal transcribed spacer of rDNA (ITS-2) has been revealed to be an important molecular marker that allows the identification of the different species of GI nematode parasites. They are used to detect polymorphism, assess cryptic species, or provide new data on the intra- and interspecific variability of species that can help in the understanding of the epidemiology, diagnosis, and control of diseases.

In this sense, the study of the structure of parasite populations allows us to determine the rates of gene flow and the risk of spread of resistance to anthelmintics among wild ruminants, which is essential if it is necessary to establish a diagnosis and treatment for any process, infectious or parasitic.

Wild animals are reservoir hosts of GI nematode species, and cross-contamination may occur from or to other domestic or unusual wild animals. An accurate taxonomy is fundamental to the identification and survey of potentially pathogenic species of parasites.

Moreover, taxonomic differentiation is used to reveal its evolution through phylogenetic studies.

Considering coprological analysis, the results obtained in the present study in roe deer revealed a mean prevalence of helminths and other parasites. GI nematodes were found with lower percentages than those found in sheep and are similar to those found in the north [66] and center of the Iberian Peninsula [61]. However, in different areas of Europe, the percentage of parasitism was considerably higher, in some cases exceeding 80% of infected animals [5,46,50,67]. In general, no study mentions the observation of clinical signs in parasitized animals. In the sheep, however, the prevalence was higher but with moderate parasite loads, similar to those found in the Salmantina meadow [49] and in central Spain [61,68], with the highest parasitic loads being found in Galicia [69].

The percentage and intensity of *Trichuris* spp. eggs found in roe deer are the highest of those cited in Spain, while in sheep they were low, similar to those observed in the northern half of the Iberian Peninsula [70,71]. Likewise, *Moniezia* spp. eggs were found in higher proportions in roe deer than in sheep. Similar values in sheep have been cited by several authors, but they were clearly higher than those found in roe deer in Toledo [61] and Galicia [56]. In Europe, various authors [5,72] reported low percentages and parasite loads for these genera in feces samples from roe deer.

There are few bibliographic references about the excretion of *Nematodirus* spp. eggs, and the existing ones refer to sheep, which are generally poorly represented, the same as in the north of Spain [69] and other European countries [73].

Coccidia oocysts in roe deer were found with a mean prevalence and low parasite load, with results similar to those found in the north of the Peninsula [12,49,71,74]. In Europe, some authors found high values in young animals [47,50,75]. These protozoa were found in practically all sheep farms with very low parasite loads. Similar data are cited in other areas of Spain [70,71,76] and somewhat lower than those obtained in the northwest of the Iberian Peninsula [12,74]. In Germany, Austria, and Greece, the prevalence depended on the age of the sheep [77,78,79], and in other areas of Europe, they were more frequent in young animals [50,72,75].

Coprological techniques are quite limited in many aspects because they depend on many factors (ecology, host, age, time of collection, or variables in the techniques), and there is no correlation between the eggs excreted by the parasites and the number of adults found within the host. However, from an epidemiological point of view, it is useful to know the parasitological status of certain hosts in a given area, and these are the techniques that are initially used in the diagnosis of any parasitic process.

## 5. Conclusions

This study has shown that the roe deer in the study area are widely infected with GI nematodes, identifying 18 species, 3 of which are polymorphic. All of them have already been cited in roe deer in Europe, but *Marshallagia marshalli*, *Capillaria bovis,* and *Ostertagia (Grosspiculopteragia) occidentalis* are reported in this host for the first time in Spain. Some species were confirmed molecularly through the analysis of the ITS-2 marker, which allows their specific identification, detects polymorphisms and cryptic species, and provides data on the variability of the species that helps us understand the epidemiology, diagnosis, and control of the diseases. Although coprological techniques are quite limited in many aspects, as there is no correlation between the eggs excreted by the parasites and the number of adults found within the host, from an epidemiological point of view, they are useful to determine their health status and are the techniques that are initially used in the diagnosis of any parasitic process.

## Figures and Tables

**Figure 1 animals-13-03117-f001:**
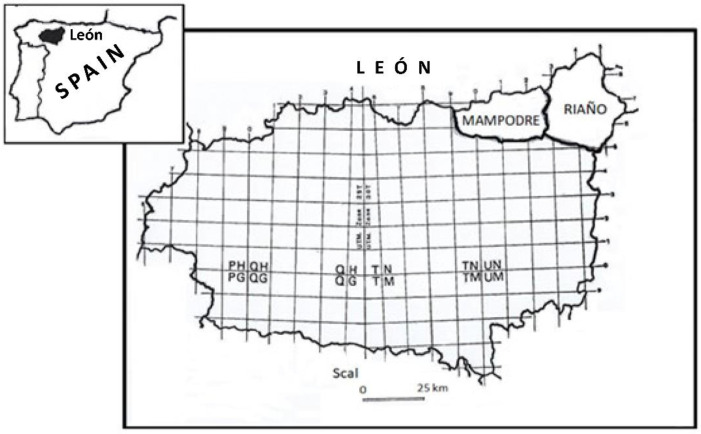
Study area. Riaño and Mampodre Regional Hunting Reserves. Province of León. Spain.

**Figure 2 animals-13-03117-f002:**
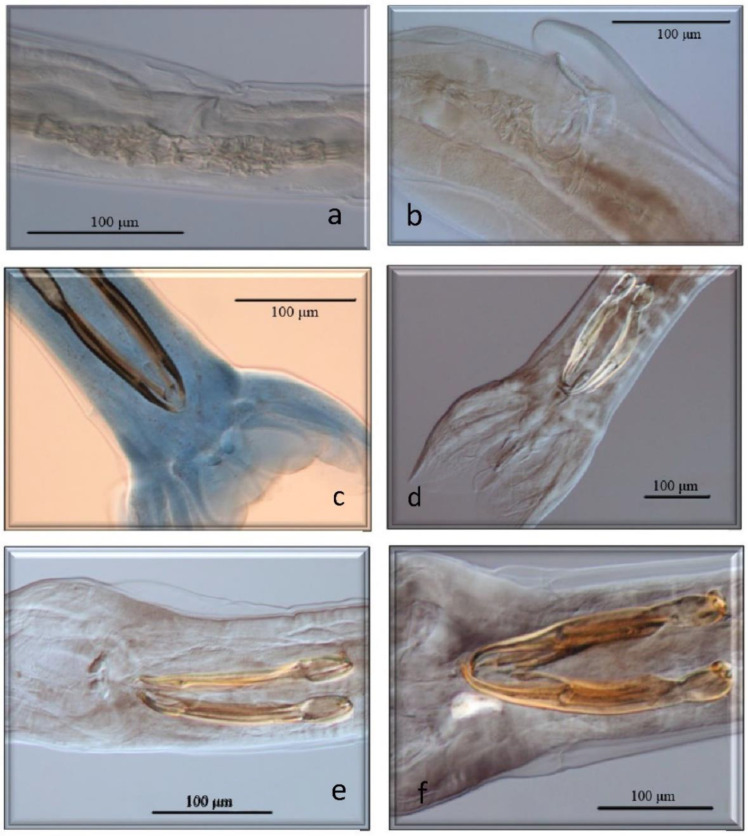
Morphological characteristics of different species of GI nematodes studied using a light microscope. (**a**) Ostertagiinae. Female without vulvar flap. (**b**) Ostertagiinae. Female with vulvar flap. (**c**) Male. *Teladorsagia circumcinta*. (**d**) Male. *Teladorsagia trifurcata*. (**e**) Male. *Ostertagia leptospicularis*. (**f**) Male. *Ostertagia kolchida*.

**Figure 3 animals-13-03117-f003:**
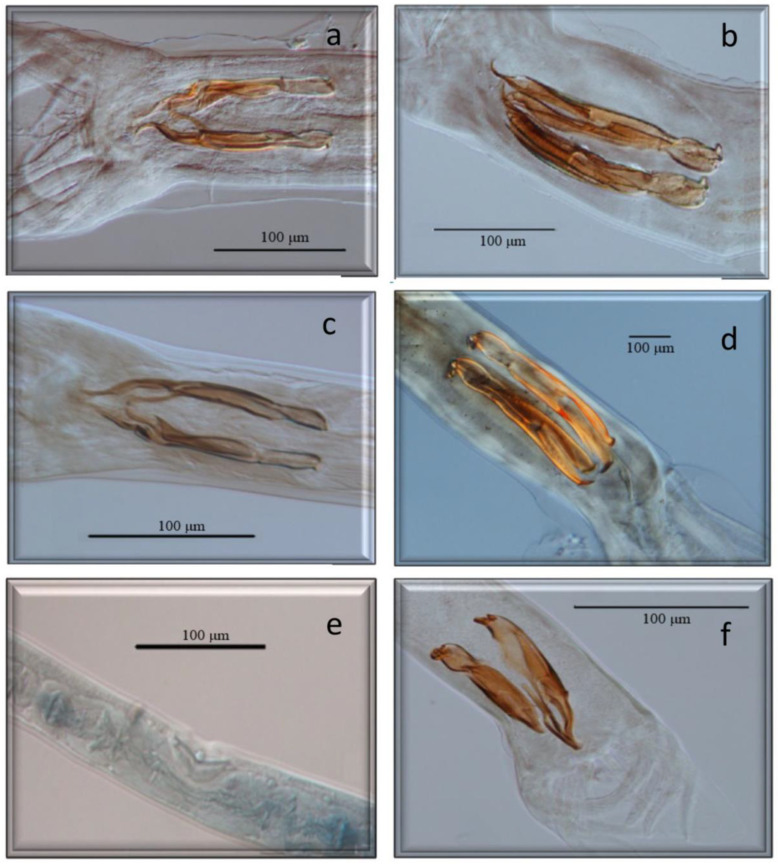
Morphological characteristics of different species of GI nematodes studied using a light microscope. (**a**) Male. *Spiculopteragia spiculoptera*. (**b**) Male. *Spiculopteragia mathewosiani*. (**c**) Male. *Marshallagia marshalli*. (**d**) Male. *Ostertagia (G) occidentalis.* (**e**) Female. *Trichostrongylus axei*. (**f**) Male. *Trichostrongylus axei*.

**Figure 4 animals-13-03117-f004:**
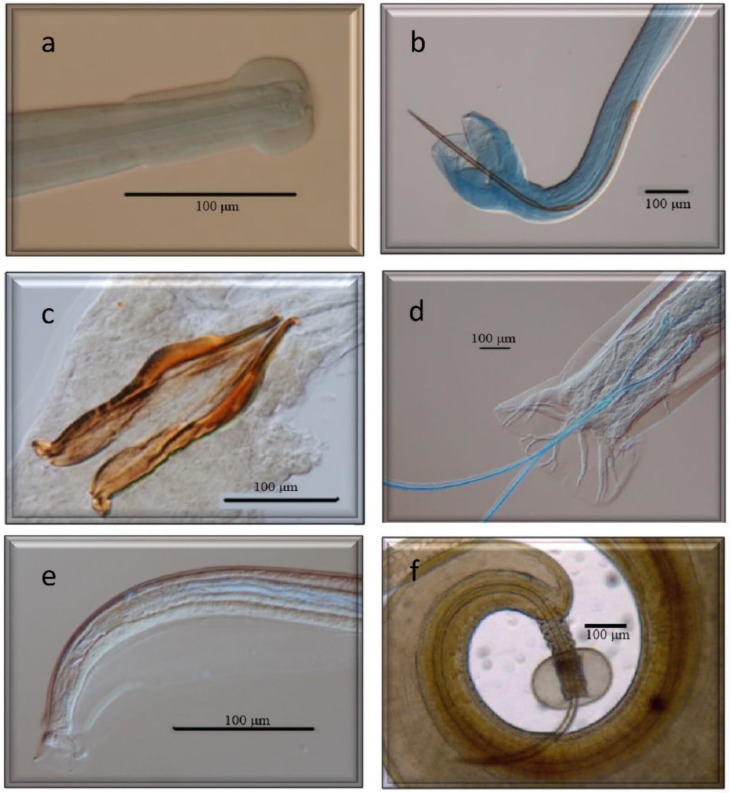
Morphological characteristics of different species of GI nematodes studied using a light microscope. (**a**) Anterior end. *Nematodirus europaeus*. (**b**) Male. *Nematodirus europaeus*. (**c**) Male. *Cooperia oncophora*. (**d**) Male. *Oesophagostomum venulosum.* (**e**) Male. *Capillaria bovis*. (**f**) Male. *Trichuris ovis*.

**Figure 5 animals-13-03117-f005:**
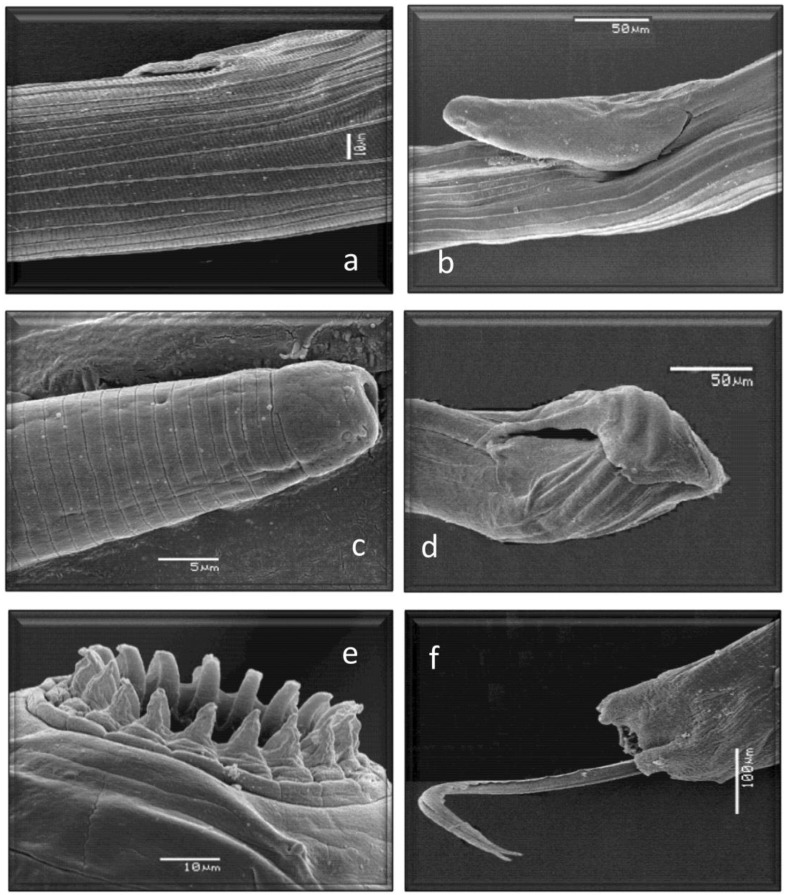
Morphological characteristics of different species of GI nematodes studied using a scanning electron microscope. (**a**) Ostertagiinae. Female without vulvar flap. (**b**) Ostertagiinae. Female with vulvar flap. (**c**) Anterior end Ostertagiinae. (**d**) Posterior end. Male Ostertagiinae. (**e**) Anterior end. Leaf crown *Oesophagostomum venulosum*. (**f**) Posterior end. Male *Oesophagostomum venulosum*.

**Figure 6 animals-13-03117-f006:**
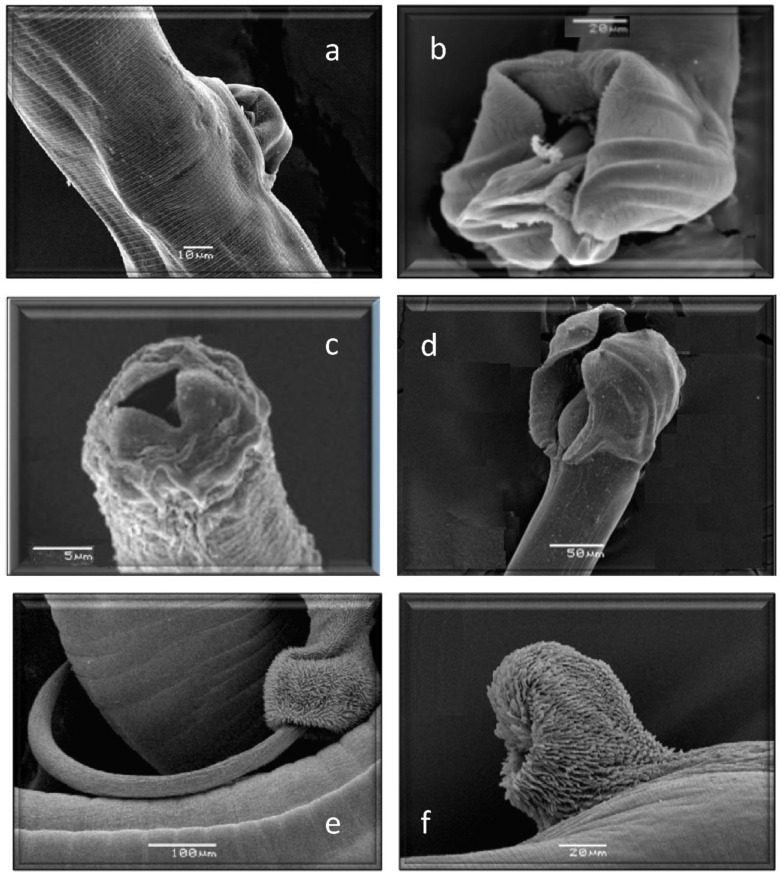
Morphological characteristics of different species of GI nematodes studied using a scanning electron microscope. (**a**) Female. *Trichostrongylus axei*. (**b**) Male. *Trichostrongylus axei*. (**c**) Anterior end. *Cooperia oncophora*. (**d**) Male. *Cooperia oncophora*. (**e**) Male. *Trichuris ovis*. (**f**) Female. *Trichuris ovis*.

**Figure 7 animals-13-03117-f007:**
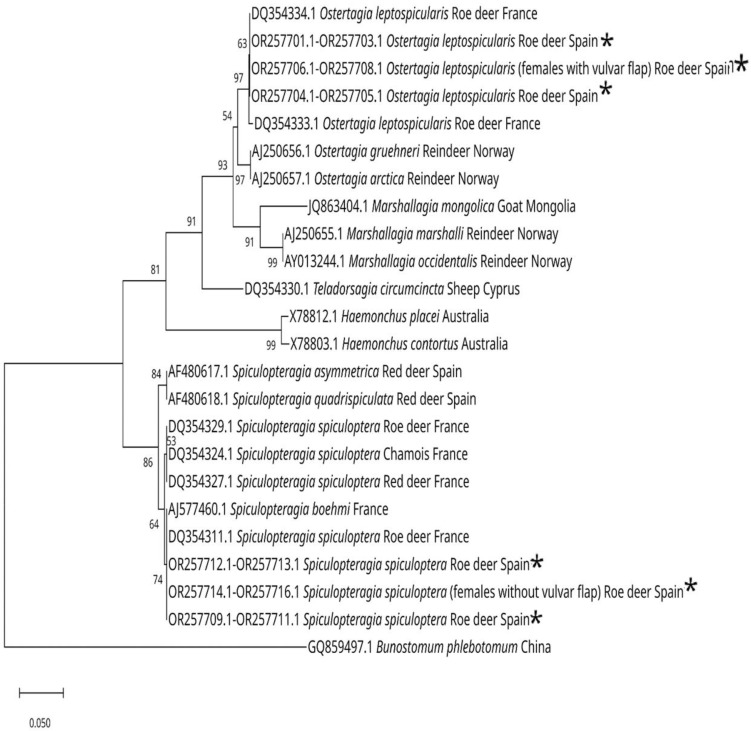
Neighbor-joining phylogenetic tree (Kimura 2-parameter model), based on sequences from the ITS-2 region, showing the position of *S. spiculoptera* and *S. mathevossiani* with *S. boehmi*, *S. aymmetrica* and *S. quadrispiculata*, *O. leptospicularis* and *O. kolchida* (in a different branch from the previous one) with *O. gruehneri* and *O. arctica* from reindeer, while *T. circumcincta* is more closely related to *M. marshalli* and *M. occidentalis*, and closer to *Ostertagia* spp. than to *Spiculopteragia* spp. [27,36,37,38,39] The species marked with an * are those obtained in this work. Bootstrap values (from 1000 replicates) for clades exceeding 50% support are mapped onto the tree. The scale shows the number of nucleotide substitutions between the DNA sequences.

**Figure 8 animals-13-03117-f008:**
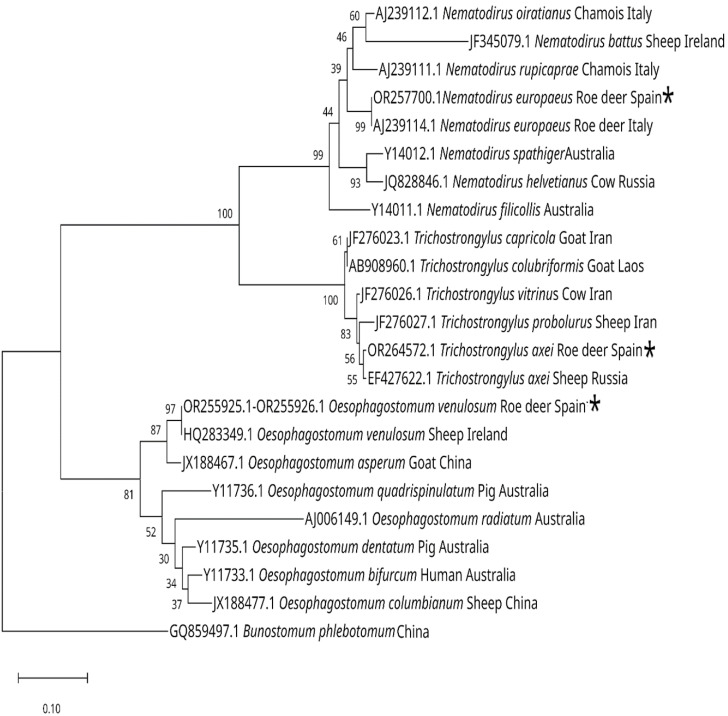
Neighbor-joining phylogenetic tree (Kimura 2-parameter model), based on sequences from the ITS-2 region, showing the position of *Nematodirus europaeus*, *Trichostrongylus axei* and *Oesophagostomum venulosum* and related species, in three different branches [29,30,40,41,42]. The species marked with an * are those obtained in this work. The percentage from 1000 replicate samples bootstrap is indicated at the right of the supported node. The scale bar shows the number of nucleotide substitutions between the DNA sequences.

**Table 1 animals-13-03117-t001:** Number of animals according to their sampling areas, age groups, and body weight.

Sampling	Host Age		Body Weight		N Roe Deer
Riaño Reserve	≤2 Years	>2 Years	≤22 kg	>22 kg	Examined
Zone A	5	13	3	15	18
Zone B	8	20	6	22	28
Zone C	14	16	10	20	30
Mampodre Reserve	5	11	7	9	16
Total	32	60	26	66	92

**Table 2 animals-13-03117-t002:** Prevalence and mean intensity of gastrointestinal nematodes found in the different parts of the digestive system of roe deer (*Capreolus capreolus*), according to their sampling areas, age groups, and body weight.

	Total		Abomasum		Small Intestine		Large Int./Cecum	
	%	$\bar{x}$ ± SD (Range)	%	$\bar{x}$ ± SD (Range)	%	$\bar{x}$ ± SD (Range)	%	$\bar{x}$ ± SD (Range)
Total n = 92	100	491.7 ± 481.0 9–2755	98.9	370.7 ± 374.4 3–1762	88.0	131.7 ± 225.6 4–1254	78.3	6.3 ± 5.5 1–26
Riaño n = 76	100	516.2 ± 504.0 23–2755	100	386.0 ± 384.7 19–1373	85.5	145.5 ± 248.6 3–1254	84.2	6.8 ± 5.6 1–23
Zone A n = 18)	100	528.8 ± 427.2 23–1413	100	394.6 ± 394.2 19–1322	88.9	141.3 ± 169.7 4–540	94.4	9.2 ± 7.1 1–23
Zone B (n = 28)	100	538.5 ± 595.0 74–2755	100	422.9 ± 394.2 38–1762	89.3	123.8 ± 209.4 3–985	75	5.0 ± 5.6 1–21
Zone C (n = 30)	100	487.7 ± 467.8 20–1102	100	346.4 ± 316.9 20–1101	80	170.9 ± 326.0 4–1254	86.7	5.4 ± 4.2 1–19
Mampodre n = 16	100	375.4 ± 340.8 9–1095	93.7	318.3 ± 322.8 3–1091	100	75.2 ± 59.9 6–179	62.5	2.9 ± 2.1 1–8
≤2 years n = 32	100	638.0 ± 646.1 9–2755	100	465.7 ± 448.3 3–1762	75	223.7 ± 362.8 1–1254	75.0	5.9 ± 6.2 1–23
>2 years n = 60	100	413.7 ± 346.0 33–1413	100	320.1 ± 321.0 21–1373	95	92.8 ± 116.2 3–540	83.3	6.4 ± 5.1 1–26
≤22 kg n = 26	100	690.8 ± 597.0 63–2755	96.1	569.0 ± 445.5 37–1762	84.6	164.4 ± 216.8 4–985	88.5	5.2 ± 4.7 1–19
>22 kg n = 66	100	413.2 ± 405.8 33–1413	100	301.3 ± 317.5 3–1322	89.4	119.4 ± 229.5 3–1107	77.3	6.8 ± 5.8 1–26

**Table 3 animals-13-03117-t003:** Prevalence and mean intensity of gastrointestinal nematode species found in the different parts of the digestive system of roe deer (*Capreolus capreolus*) (A = abomasum; SI = small intestine).

Species (18 spp.)	Abomasum		Small Intestine		Large Int./ Cecum	
	%	$\bar{x}$ ± SD (Range)	%	$\bar{x}$ ± SD (Range)	%	$\bar{x}$ ± SD (Range)
*Teladorsagia circumcincta* n = 78	84.8	47.9 ± 68.3 (1–400)				
*Teladorsagia trifurcata * n = 19	20.6	3.6 ± 3.9 (1–13)				
*T. circumcincta/T. trifurcata * n = 80	86.9	47.6 ± 67.9 (1–400)				
*Spiculopteragia spiculoptera * n = 69	75	42.4 ± 61.9 (1–365)				
*Spiculopteragia mathevossiani * n = 11	11.9	10.8 ± 21.7 (1–73)				
*S. spiculoptera/S. mathevosiani * n = 71	77.2	42.9 ± 62.6 (1–365)				
*Ostertagia leptospicularis * n = 11	11.9	14.9 ± 11.1 (2–28)				
*Ostertagia kolchida * n = 42	45.7	9.9 ± 15 (1–84)				
*O. leptospicularis/O. kolchida * n = 49	53.3	11.6 ± 14.9 (1–84)				
*Marshallagia marshalli * n = 29	31.5	6.6 ± 6.8 (1–18)				
*O. (Grospiculopteragia) occidentalis * n = 4	4.4	3.3 ± 2.1 (1–6)				
*Haemonchus contortus * n = 15	16.3	5.5 ± 7.0 (1–29)				
*Trichostrongylus axei * n = 78 (A); n = 57 (SI)	84.8	52.9 ± 74.2 (1–336)	51,2	5.5 ± 5.5 (1–46)		
*Trichostrongylus vitinus * n = 32 (A); n = 22 (SI)	34.8	4 ± 4.7 (1–27)	23.9	4.5 ± 6.2 (1–29)		
*Trichostrongylus capricola * n = 15 (A); n = 19 (SI)	16.3	1.9 ± 1.5 (1–7)	20.6	3.6 ± 5.4 (1–24)		
*Trichostrongylus colubriformis * n = 9 (A); n = 9 (SI)	9.8	2.3 ± 1.9 (1–6)	9.8	3.4 ± 2.8 (1–8)		
*Nematodirus europaeus * n = 7 (A); n = 74 (SI)	7.6	10.7 ± 11 (1–7)	80.4	115.3 ± 221.7 (1–1226)		
*Capillaria bovis * n = 33			35.9	5 ± 6.8 (1–27)		
*Cooperia oncophora * n = 16			17.4	6.6 ± 11.1 (1–46)		
*Trichuris ovis * n = 62					64.1	3.7 ± 2.2 (1–10)
*Oesophagostomum venulosum * n = 54					58.7	4.3 ± 4.4 (1–19)

**Table 4 animals-13-03117-t004:** Results of coprological analysis.

	Total Sheep n = 151	Total Roe Deer n = 92	Mampodre Roe Deer n = 16	Riaño Roe Deer n = 76
Trichostrongylid eggs	87.7% $\bar{x}$ = 143 ± 233 (13–1.700)	65.2% $\bar{x}$ = 47.3 ± 46.8 (6–260)	75% $\bar{x}$ = 32.9 ± 29.2 (13–100)	63.2% $\bar{x}$ = 32.0 ± 46.5 (12–260)
*Nematodirus* spp.	21,2% $\bar{x}$ = 42 ± 26 (13–186)	––––––––	–––––––	–––––––
*Trichuris* spp.	3.3% $\bar{x}$ = 20 ± 4 (13–46)	52.2% $\bar{x}$ = 19.1 ± 17 (6–66)	18.7% $\bar{x}$ = 13 ± 0 (13–13)	59.2% $\bar{x}$ = 19.1 ± 16.5 (6–33)
*Moniezia* sp.	6%	21.7%	37.5%	18.4%
Coccidia oocysts	96% $\bar{x}$ = 89 ± 278 (13–2500)	49% $\bar{x}$ = 574 ± 2847	37.5% $\bar{x}$ = 3295 ± 779 (50–19200	51.3% $\bar{x}$ = 156 ± 211 (13–800)

## Data Availability

The data that support the findings of this study are available in GenBank at https://www.ncbi.nlm.nih.gov/genbank/ (accessed on 27 September 2023), with the accession numbers: OR257701, OR257702, OR257703, OR257704, OR257705, OR257706, OR257707, OR257708, OR257709, OR257710, OR257711, OR257712, OR257713, OR257714, OR257715 and OR257716, OR264572, OR257700, OR255925 and OR255926.

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
