# Peer review of "Contribution to the Knowledge of Gastrointestinal Nematodes in Roe Deer (Capreolus capreolus) from the Province of León, Spain: An Epidemiological and Molecular Study"

_animals, 2023, doi:10.3390/ani13193117_

Round 1

Reviewer 1 Report

The manuscript entitled „Contribution to Knowledge of Gastrointestinal Nematodes in

Roe Deer (Capreolus capreolus) from the Province of León, Spain: Study Epidemiological and Molecular” describes interesting and original work. Roe deer is one of the most common wild animals in Europe thus examination of its parasitic infections and their transmission to livestock is an important research topic. The study described in the manuscript also provides new data in the area of parasitic infections of roe deer.

The research was well planned and conducted, however the manuscript, especially discussion, need some changes. Firstly, English used in the manuscript demands an improvement by a native speaker.

Simple summary

Line 13 – Please do not use the word „shedding” instead of „elimination”.

Line 13-15 – I would suggest to move this part (about examination of sheep and roe deer faeces) AT the end of the abstrakt. It was not the main part of the study and it should be presented AT the end of the abstrakt, after results of examination of nematodes.  

Please correct the same in the abstract.

Introduction

Line 59-62 – The citation is missing

Line 66-68 – also lack of the citation

Line 69-71 – this paragraph about gastrointestinal nematodes of roe deer should be expanded. It is too brief.

Line 75 – “pastures or watering holes”

Line 80-86 – English demands multiple corrections. In the first sentence the authors did not even mention that it is about roe deer parasites. Please use “parasitic infections” instead of “parasitization”. The whole last paragraph of the introduction is one sentence. Please divide it on at least two sentences.

Materials and methods - I would like to suggest the authors to describe collection of faecal samples at the end of materials and methods, before statistical analysis and after molecular analysis. Examination of faecal samples should be also described. Please provide it.

Results

Line 194-209 – I would suggest to move this paragraph at the end of the results section. Examination of roe deer and sheep faeces was not the main part of this study. Results of the dissections of roe deer are more important and reliable than faecal examination. Additionally, as describing examination of faeces, please start from roe deer and compare it to sheep, not the other way round.   

Line 205 – please use “shedding” instead of “elimination”.

Line 211 – It should be: “The prevalence of nematodes in the gastrointestinal tract of roe deer examined in the present study was...” .

Line 215 – It should be “in 2 young animals it was” instead of “was it”

Line 331 – it should be “the abomasum was the part of gastrointestinal tract”

Line 363 – it should be “S. spiculoptera” instead of “S.spiculopteragia

Line 385 – It should be “Parasites were identified in 88% (...) of the small intestines of examined animals...”

Tables:

Table 2 –Capillaria spp. should not be listed in the table because it was not present in the examined samples.

Table 3 – it should be “range” instead of “rango”. The authors could also try to broaden some columns in the table to make it more visible.

Figures:

Most of figures with photographs of nematodes can be grouped together as only few multi-photo figures instead of 30 separate photographs.

Figure 32 and 33 – please include an information about number of generations the analysis was run for. I would also like to ask what is an outgroup of both phylogenetic trees and why some of the support scores are so low, like 29, 15 or 9. I am not sure if it should be included in the tree.

 Discussion

The discussion is poorly written, with no major conclusions. It is mostly based on citing other studies without any conclusions about present ones. Once again I believe that examination of roe deer faeces  and comparison the results with faecal examination of domestic sheep should be discussed at the end, not the beginning of the discussion. Additionally, as discussing examination of faeces, please start from roe deer and compare it to sheep, not the other way round.  Examination of faeces is very limited method and it does not necessarily allow to draw reliable conclusions, so the authors should not be so focused on this part of the study.  This limitation should be also mentioned in the discussion.

The main part of the present study was examination of gastrointestinal nematodes isolated during roe deer dissections. General prevalence and intensity of parasitic infections in roe deer is also poorly discussed. Was it higher than in other studies in Spain and Europe?

The results that roe deer with smaller weight had higher parasite loads might be connected to the fact that young animals were more parasitized. Thus one fact might be a result of the other. It is worth mentioning.  

Please underline the importance of molecular examination, provided in the present study. It brings new knowledge and value to the research of gastrointestinal nematodes.

There should be a separate paragraph at the end of the discussion with few major conclusions.

Nevertheless, the manuscript presents very interesting work and I recommend it for publication after the major revision.

The language should be improved. I suggest the manuscript should be checked by the native speaker.

Author Response

REVIEWER 1

Simple summary

Line 13 – Please do not use the word „shedding” instead of „elimination”.

- We think he is wrong. Ok, we will use shedding instead of elimination

Line 13-15 – I would suggest to move this part (about examination of sheep and roe deer faeces) AT the end of the abstrakt. It was not the main part of the study and it should be presented AT the end of the abstrakt, after results of examination of nematodes. 

Please correct the same in the abstract.

- Ok, it moves where you indicate

Introduction

Line 59-62 – The citation is missing

- Do you really need a bibliographical reference to say that roe deer live in the northern hemisphere???

Line 66-68 – also lack of the citation

- There is no appointment, it is a personal communication

Line 69-71 – this paragraph about gastrointestinal nematodes of roe deer should be expanded. It is too brief.

- Although the paragraph is short, we believe that it is not necessary to expand it. The species cited in Europe and Spain can be consulted in the cited bibliographical references.

Line 75 – “pastures or watering holes”

- Ok, sorry

Line 80-86 – English demands multiple corrections. In the first sentence the authors did not even mention that it is about roe deer parasites. Please use “parasitic infections” instead of “parasitization”. The whole last paragraph of the introduction is one sentence. Please divide it on at least two sentences.

 - The roe deer is a free-living animal and is an ungulate and as such we refer to it. In the first sentence it is said that we want to know what species affect this ungulate, if we are talking about the roe deer, what are we referring to? It is poorly expressed or poorly written and that is why the English needs multiple corrections??? We don't understand this comment

We will use parasitic infections and the last paragraph has been divided into two sentences

Materials and methods - I would like to suggest the authors to describe collection of faecal samples at the end of materials and methods, before statistical analysis and after molecular analysis. Examination of faecal samples should be also described. Please provide it.

 - Ok

Results

Line 194-209 – I would suggest to move this paragraph at the end of the results section. Examination of roe deer and sheep faeces was not the main part of this study. Results of the dissections of roe deer are more important and reliable than faecal examination. Additionally, as describing examination of faeces, please start from roe deer and compare it to sheep, not the other way round.

- As previously stated, it has been moved and rewritten with your indications. 

Line 205 – please use “shedding” instead of “elimination”.

- Ok already changed

Line 211 – It should be: “The prevalence of nematodes in the gastrointestinal tract of roe deer examined in the present study was...” .

- Ok already changed

Line 215 – It should be “in 2 young animals it was” instead of “was it”

- Ok already changed

Line 331 – it should be “the abomasum was the part of gastrointestinal tract”

- Ok already changed

Line 363 – it should be “S. spiculoptera” instead of “S.spiculopteragia”

- Ok already changed

Line 385 – It should be “Parasites were identified in 88% (...) of the small intestines of examined animals...”

 - Ok already changed

Tables:

Table 2 –Capillaria spp. should not be listed in the table because it was not present in the examined samples.

- Ok has been deleted

Table 3 – it should be “range” instead of “rango”. The authors could also try to broaden some columns in the table to make it more visible.

- Sorry, “rango” we did not translate it into English. We tried to expand the columns but there is not much space

Figures:

Most of figures with photographs of nematodes can be grouped together as only few multi-photo figures instead of 30 separate photographs.

- In the original manuscript the figures were sent in a composition with six photographs per figure and we believed that the publisher would keep it as they were sent. In this revision the figures are reduced to five, each of them composed of the 6 initial photographs, numbered with a,b,c,d,e,f,g and sent as TIFF images.

Figure 32 and 33 – please include an information about number of generations the analysis was run for. I would also like to ask what is an outgroup of both phylogenetic trees and why some of the support scores are so low, like 29, 15 or 9. I am not sure if it should be included in the tree.

- The trees have again been revised and reformed. The number of generations for the realization of the trees was 1000 and has been added in the epigraph of the figures. The external group used was the species Bunostomum flebotomum and the support scores are so low were due to the fact that in the GenBank there are few sequences available for these species.

Discussion

The discussion is poorly written, with no major conclusions. It is mostly based on citing other studies without any conclusions about present ones. Once again, I believe that examination of roe deer faeces and comparison the results with faecal examination of domestic sheep should be discussed at the end, not the beginning of the discussion. Additionally, as discussing examination of faeces, please start from roe deer and compare it to sheep, not the other way round. 

Examination of faeces is very limited method and it does not necessarily allow to draw reliable conclusions, so the authors should not be so focused on this part of the study.  This limitation should be also mentioned in the discussion.

- The discussion is rewritten.

- Obviously, the coprological data would be placed at the end of the study, in accordance with the previous indications.

The main part of the present study was examination of gastrointestinal nematodes isolated during roe deer dissections. General prevalence and intensity of parasitic infections in roe deer is also poorly discussed. Was it higher than in other studies in Spain and Europe?

- In terms of prevalence, it is one of the highest results but not so in terms of the average intensity of infection.

These results will depend on the ecological conditions, favourable or not, for the development of the free phases of the biological cycle of the parasites, which will be different depending on the geographical area studied (paragraph added to the discussion).

The results that roe deer with smaller weight had higher parasite loads might be connected to the fact that young animals were more parasitized. Thus one fact might be a result of the other. It is worth mentioning. 

- Already explained in the text

Please underline the importance of molecular examination, provided in the present study. It brings new knowledge and value to the research of gastrointestinal nematodes.

- Introduced in the text a series of considerations on the molecular study

There should be a separate paragraph at the end of the discussion with few major conclusions.

 - Ok

Reviewer 2 Report

This is an interesting article which builds on limited knowledge on the prevalence and species of parasites in wild animals- in this case roe deer. It is well written in the main and I only have a few comments largely on grammar which are detailed below with a suggestion for modifications.

I would like a bit more details on the PCR methodology. Also the images are excellent, but there are a lot of them, so the authors may wish to consider grouping some together. But that is up to their discretion.

Line 42 - d than those in better physical condition … (reword)

Line 63- please define M/H ratio

Line 69- Roe deer host a variety …. (reword)

Line 72- the importance of GI nematodes …. (reword)

Line 73- of the same parasites and …. (reword)

Line 75- pasture or watering holes…. (reword)

Line 80-86- this needs splitting into two or more sentences for ease of reading

Line 159- a lysis solution was used (reword)

Line 173- some detail on PCR reagents and cycling conditions maybe useful here?

Table 2- I think this ‘Ooquiste Coccidios’ should be Coccidia oocysts?

Line 228- could this be selection? Animals with higher parasites are less fit, so more likely to be shot?

Line 237-247- this is quite difficult to read and digest. Is there a way to present it slightly clearer?

Figures 2-31-  I think you have some excellent images of the worms here which should be commended. However, I do wonder if all are requires, or if having some in a block, e.g. Figure 2a, 2b and 2c for example may make it  a bit more reader friendly? Just a suggestion but up to the authors

Figure 32 and 33- the * is hard to see. Could you consider bolding the species in the tree for clarity?

Line 343- parasite load than those in better physical condition….(reword)

Line 370 - Ostertagia (Grosspiculopteragia) occidentalis is reported for the first time in this host in …. (reword)

Line 398-403- please put Latin names into italics. And line 554

Line 441- is this single infections rather than simple ones?

Line 463- presented with a vulvar flap …. (Reword)

Line 452-489- this gets rather repetitive and isn’t really results. It would be nicer to see a comparison of the species, and why they are different and where, and what that means rather than just say that they were compared

Section 3.4. This isn’t really needed as it doesn’t add much

Line 542- in domestic animals… (reword)

Line 592- that presented with a vulval flap …. (reword)

Line 596- what do you mean with her? Please make clearer and reword

Line 605- wrong referencing style

These are detailed above

Author Response

REVIEWER 2

Line 42 - d than those in better physical condition … (reword)

“Animals with lower body weight had a higher parasite load than those with better physical condition, and in this case, statistically significant differences were found” It has been changed to the following sentence, but we don't really know what it means to rewrite, for us it is well written

Line 63- please define M/H ratio

- Male/Female. Sorry

Line 69- Roe deer host a variety …. (reword)

- The roe deer presents a wide variety ....(????)

Line 72- the importance of GI nematodes …. (reword)

- Sorry, we think it is perfectly expressed

Line 73- of the same parasites and …. (reword)

- “of the same parasites”  and  the following we believe is well expressed

Line 75- pasture or watering holes…. (reword)

- Sorry, what do we have to reword ????????

Line 80-86- this needs splitting into two or more sentences for ease of reading

- Ok, already divided into two paragraphs

Line 159- a lysis solution was used (reword)

- This has been changed as you indicate

Line 173- some detail on PCR reagents and cycling conditions maybe useful here?

- If you consider it really necessary we would introduce how we perform the PCR but they are standardized protocols and we believe that it does not make much sense

Table 2- I think this ‘Ooquiste Coccidios’ should be Coccidia oocysts?

- Ok, sorry

Line 228- could this be selection? Animals with higher parasites are less fit, so more likely to be shot?

- Obviously it can be a selection process, parasitized animals, weaker, without a great ability to escape, can be more easily shot by hunters and can be easy prey for their predators.

Line 237-247- this is quite difficult to read and digest. Is there a way to present it slightly clearer?

- Ok, we'll try

Figures 2-31-  I think you have some excellent images of the worms here which should be commended. However, I do wonder if all are requires, or if having some in a block, e.g. Figure 2a, 2b and 2c for example may make it  a bit more reader friendly? Just a suggestion but up to the authors

- In the original manuscript the figures were sent in a composition with six photographs per figure and we believed that the publisher would keep it as they were sent. In this revision the figures are reduced to five, each of them composed of the 6 initial photographs, numbered with a, b, c, d, e, f, g and sent as TIFF images.

Figure 32 and 33- the * is hard to see. Could you consider bolding the species in the tree for clarity?

- The trees have been renovated due to some doubts expressed about them but we have forgotten to put the species in bold. We're sorry. We think they look better now

Line 343- parasite load than those in better physical condition….(reword)

- The same as in line 42. We do not understand very well what you want us to reform

Line 370 - Ostertagia (Grosspiculopteragia) occidentalis is reported for the first time in this host in …. (reword)

- Ostertagia (Grosspiculopteragia) occidentalis are reported in this host for the first time in Spain???

Line 398-403- please put Latin names into italics. And line 554

- Ok, sorry

Line 441- is this single infections rather than simple ones?

- Single infections in Parasitology are called simple

Line 463- presented with a vulvar flap …. (Reword)

- The vulvar flap is a characteristic of the females of some nematodes whose vulva is covered by a finger-shaped swelling of the cuticle. It has no other name as can be observed in all parasitology books

Line 452-489- this gets rather repetitive and isn’t really results. It would be nicer to see a comparison of the species, and why they are different and where, and what that means rather than just say that they were compared

- Molecular identification is not done like this? The sequence deposited in GenBank of a certain species is obtained and it is compared and aligned with the sequence obtained by us. If both sequences match, they are the same species.

Section 3.4. This isn’t really needed as it doesn’t add much

- This is quite debatable

Line 542- in domestic animals… (reword)

- Animals have been added

Line 592- that presented with a vulval flap …. (reword)

- Same as line 463

Line 596- what do you mean with her? Please make clearer and reword

- We are referring to the vulvar flap, some species do not have a flap and others were observed with this characteristic. Sorry

Line 605- wrong referencing style

- Ok, sorry

Yours sincerely,

Natividad Díez Baños

Prof. Dra. Natividad Díez Baños

Dpto. Sanidad Animal (Parasitología y Enfermedades Parasitarias)

Facultad de Veterinaria. Universidad de León

Campus de Vegazana s/n

  1. León. España

Phone:+34-987-291333

Fax: +34-987-291304

E-mail: mndieb@unileon.es